# Efficacy and Safety of the Genistein Nutraceutical Product Containing Vitamin E, Vitamin B3, and Ceramide on Skin Health in Postmenopausal Women: A Randomized, Double-Blind, Placebo-Controlled Clinical Trial

**DOI:** 10.3390/jcm12041326

**Published:** 2023-02-07

**Authors:** Mingkwan Na Takuathung, Preeyaporn Klinjan, Wannachai Sakuludomkan, Nahathai Dukaew, Ratchanon Inpan, Rattana Kongta, Wantida Chaiyana, Supanimit Teekachunhatean, Nut Koonrungsesomboon

**Affiliations:** 1Department of Pharmacology, Faculty of Medicine, Chiang Mai University, Chiang Mai 50200, Thailand; 2Clinical Research Center for Food and Herbal Product Trials and Development (CR-FAH), Faculty of Medicine, Chiang Mai University, Chiang Mai 50200, Thailand; 3Research Center of Pharmaceutical Nanotechnology, Faculty of Pharmacy, Chiang Mai University, Chiang Mai 50200, Thailand; 4Department of Pharmaceutical Sciences, Faculty of Pharmacy, Chiang Mai University, Chiang Mai 50200, Thailand; 5Musculoskeletal Science and Translational Research Center, Faculty of Medicine, Chiang Mai University, Chiang Mai 50200, Thailand

**Keywords:** genistein, menopause, nutraceuticals, postmenopausal women, randomized-controlled trial, skin aging

## Abstract

Skin aging is one of the most concerning issues that occur after menopause. The Genistein Nutraceutical (GEN) product, containing genistein, vitamin E, vitamin B3, and ceramide, has been formulated as a topical anti-aging product for improving the health of postmenopausal women’s facial skin. This study aimed to investigate the efficacy and safety of the GEN product on postmenopausal women’s facial skin. This randomized, double-blind, placebo-controlled trial randomly assigned 50 postmenopausal women to receive either the GEN product (n = 25) or the placebo (PLA) product (n = 25), topically applied twice daily for 6 weeks. The outcome assessments included multiple skin parameters related to skin wrinkling, color, hydration, and facial skin quality at baseline and week 6. The percentage mean changes or absolute mean changes, where appropriate, in skin parameters were compared between the two groups. The mean age of the participants was 55.8 ± 3.4 years. For skin wrinkling and skin color parameters, only skin redness was significantly higher in the GEN group when compared to the PLA group. Following the application of the GEN product, skin hydration increased while fine pores and their area decreased. Subgroup analysis of older women (age ≥ 56 years) with adequate compliance found significant differences between the two groups in the percentage mean changes of most skin wrinkle parameters. The GEN product has benefits for the facial skin of postmenopausal women, particularly those who are older. It can moisturize facial skin, lessen wrinkles, and enhance redness.

## 1. Introduction

In women, estrogen naturally declines after menopause, and several physiologic changes occur due to low estrogen levels [1]. Skin is one of the target organs where estrogens have a substantial impact on the aging process; it undergoes significant changes after menopause, such as elasticity loss, increased wrinkles and dryness, decreased epidermal thickness and collagen content, and the degeneration of elastic fiber [2,3]. As a result of this, postmenopausal women may experience a deterioration in their perceived facial beauty [4]. The annual market for skincare products that promise to slow down or delay the aging process of facial skin in postmenopausal women was valued at USD 15.4 billion in 2021 and is expected to continue growing, demonstrating the importance of this issue [5,6,7]. It is, therefore, of interest to further develop innovative skincare products that can serve postmenopausal women in particular.

Phytoestrogens, or plant-derived compounds that mimic estrogen in the human body, can exert an anti-aging effect on the skin via estrogen receptors [8]. Among them, genistein is one of the most active phytoestrogens, primarily found in soybeans, and exerts its skincare properties via the estrogen receptor beta (ERβ) [9]. Clinical evidence suggests the benefit of topical therapy with genistein in improving facial wrinkles, skin dryness, and fibroblast viability, as well as in increasing hyaluronic acid levels and type I and III collagen production [10]. Therefore, genistein-based nutraceutical products are becoming one of the high-potential product innovations that can satisfy postmenopausal women’s needs thanks to their anti-aging effects on the skin. The development of innovative skincare products based on the use of natural genistein-based extracts will also add economic value to medicinal plants, particularly soybeans.

The Genistein Nutraceutical (GEN) product, which contains genistein and some other ingredients, including vitamin E, vitamin B3, and ceramide, has been formulated as a topical alternative to anti-aging products for improving the health of postmenopausal women’s facial skin. Apart from the beneficial properties of genistein on the facial skin, it is well known that vitamin E, vitamin B3, and ceramide also have a significant impact on the improvement of facial skin conditions. The topical application of vitamin E is able to alter the biosynthesis of collagen and glycosaminoglycan in the skin owing to its antioxidant properties [11,12]. Vitamin B3 (also known as niacinamide or nicotinamide) in a topical form can exert a stabilizing effect on epidermal barrier function, presenting with a decrease in transepidermal water loss and an increase in the moisture content of the stratum corneum [13]. Besides, it can increase the biosynthesis of ceramide and other stratum corneum lipids to improve the epidermal permeability barrier [14]. Ceramides, which are sphingolipids presenting in the intercellular space of the stratum corneum, play an essential role in structuring and maintaining the barrier property of the epidermis and exhibit a certain level of efficacy in improving skin dryness [15,16]. Although there are several in vitro and in vivo studies on the benefits of genistein [17], vitamin E [18], vitamin B3 [19], and ceramide [20] alone, limited studies have been conducted to assess the effects of their combined regimen on facial skin.

The present study aimed to assess the efficacy of the GEN product on postmenopausal women’s facial skin by means of a randomized, double-blind, placebo-controlled design. It was also designed to assess the product’s safety and participant satisfaction.

## 2. Materials and Methods

### 2.1. Study Design and Setting

This randomized, double-blind, placebo-controlled trial was conducted at the Clinical Research Center for Food and Herbal Product Trials and Development (CR-FAH), Faculty of Medicine, Chiang Mai University, Chiang Mai, Thailand, between May and July 2022. The clinical trial protocol and supporting documents were approved by the Research Ethics Committee of the Faculty of Medicine, Chiang Mai University (No. 045/2022). This study was prospectively registered with the Thai Clinical Trials Registry (TCTR20220209010) prior to enrollment.

### 2.2. Study Participants and Sample Size Determination

Postmenopausal women (aged between 48 and 65 years) who had stopped menstruation for at least 12 months were eligible for this study. Other inclusion criteria were as follows: a serum level of follicle-stimulating hormone (FSH) of >30 IU/L, a body mass index between 18 and 30 kg/m^2^, no significant laboratory abnormalities (including a complete blood count, aspartate transaminase, alanine aminotransferase, alkaline phosphatase, blood urea nitrogen, creatinine, and serum electrolytes), and the ability to read and write Thai and give written informed consent. Those who (1) had used estrogen or isoflavones within the six months prior to study entry, (2) had a history of allergy to soy products, vitamin E, vitamin B3, or ceramides, (3) had an underlying disease for which soy products were contraindicated (such as skin cancer or skin infection, or any chronic diseases), (4) used medications (e.g., antibiotics, steroids, and immunosuppressants), food supplements, herbs, vitamins, minerals, or other products that might significantly affect the skin within the three months prior to study entry, and (5) used narcotics or psychotropic substances, consumed excessive amounts of alcohol, or smoked cigarettes on a regular basis were excluded from this study. All the participants signed a written informed consent form prior to study entry.

This trial planned to enroll a total of 50 participants (25 in each group). The sample size of 25 per group was estimated to achieve a power of 80% and a level of significance of 5% (two-sided) for detecting an effect size of 0.9 with a dropout rate of 10%.

### 2.3. Study Intervention and Comparator

Genistein was purchased from Xi’an Rongsheng Biotechnology Co., Ltd. (Xi’an, China), with a purity of 99.03% (*w*/*w*) as analyzed by high-performance liquid chromatography coupled with a diode array detector and a mass spectrometric detector (HPLC-DAD-MSD, Central Laboratory (Thailand) Co., Ltd., Chiang Mai, Thailand). The Genistein Nutraceutical (GEN) product consisted of 4% genistein, 1% vitamin E, 1% vitamin B3, and 0.2% ceramide. The placebo (PLA) product had a cream base without genistein, vitamin E, vitamin B3, or ceramide; it was packed in an opaque bottle identical to the GEN product.

### 2.4. Randomization, Blinding, and Allocation Concealment

Before trial initiation, computer-generated random numbers were acquired, and a 1:1 allocation utilizing a block size of 10 was used. Research staff who were not involved in the study’s clinical aspects dispensed either GEN or PLA to the participants. The investigator team was not made aware of the allocation sequence by means of sequentially numbered, opaque, sealed envelopes. Only after each participant had met the inclusion/exclusion criteria were the corresponding envelopes opened. Investigators, including outcome assessors, were kept blinded to the group assignment of each participant.

### 2.5. Study Procedures

An occlusive patch test was used on the participants, in which the product was applied to a 3 × 3 cm area of the forearm and covered with tape for 24–48 h to assess any allergic reactions. If some participants developed allergic rashes, they would be withdrawn from the study. In such cases, new volunteers would be recruited to replace them.

At baseline, the participants underwent an assessment of their facial skin condition. After that, they were given a product (based on the randomization list) and were instructed to apply the product to their facial skin twice daily (0.6 mL in the morning and another 0.6 mL at night) for 6 weeks. The participants were instructed not to use any other cosmetics over the course of the trial, and they were also forbidden from ingesting any herbs or other nutritional supplements that were known to affect their skin condition.

### 2.6. Outcome Assessment

Outcome assessment was performed at baseline and at the end of the study. Efficacy outcome measures included five sets of skin parameters, as follows:(1)Skin wrinkling parameters, including depth of skin roughness (R1, the distance between the highest and the lowest value), maximum roughness (R2, the biggest roughness of the different segment roughness values), average roughness (R3, the arithmetic average of the different segment roughness values), smooth depth (R4, the average distance between the real profile and the maximum profile), arithmetic average roughness (R5, the distance between the average line and the average deviation), surface (the size of the wavy surface which was compared to a fully stretched flat surface), and volume (the virtual amount of liquid needed in the calculation area to fill the image until the average height of all mountains), all of which were measured by the Skin Visiometer SV 700 (Courage and Khazaka electronic, Cologne, Germany);(2)Skin color parameters, including brightness (LAB1, L-skin color), redness (LAB2, A-the dermatologist’s perception of skin redness and erythema), pigmentation (LAB3, B-pigmentation, and tanning), and individual typology angle (ITA, skin color-white-dark whitening), all of which were measured by the Skin-Colorimeter^®^ CL 400 (Courage and Khazaka electronic, Cologne, Germany);(3)Hydration value, which was measured by the Corneometer (Cutometer^®^ dual MPA 580, Courage and Khazaka electronic, Cologne, Germany);(4)Facial skin parameters, including fine pores, large pores, and spots, all of which were measured by the Visioface 1000 D (Courage and Khazaka electronic, Cologne, Germany); and(5)Acne parameters, including size and quantity, were all measured by the Visiopor^®^ PP34 (Courage and Khazaka electronic, Cologne, Germany) (using a specific UV light to visualize the fluorescing acne lesions).

Five dimensions of participant satisfaction (serum formulation, wrinkle reduction, hydration, overall efficacy, safety profile, and whether to recommend the product to friends and family) were assessed at the end of the study using the 10-point scale questionnaire, with a higher value indicating better satisfaction. Any adverse events seen by the investigators or self-reported by the participants were recorded. The weight of the product remaining in the bottle was measured to determine compliance. Participants who used the product at less than 50% of the prescribed dosage were considered poorly compliant.

### 2.7. Statistical Analysis

The intention-to-treat (ITT) and per-protocol (PP) approaches were used to assess the outcomes of this study. All participants who were randomly allocated were included in the ITT method, and the analysis was based on the group to which they were initially assigned. The analysis utilizing the PP approach excluded participants with poor compliance or loss of follow-up. Continuous variables are presented as mean ± standard deviation (SD). Percentage mean changes from baseline or absolute mean changes from baseline between the two groups were compared using the Student’s *t*-test; the mean difference (MD) between the two groups and its 95% confidence interval (95% CI) were then obtained. Subgroup analysis was performed to determine the effect of the product among postmenopausal women aged 56 years or older who had applied the product with adequate compliance.

All statistical analyses were performed using IBM SPSS Statistics for Windows, Version 22.0. Armonk, NY, USA: IBM Corp., Release 2013. A *p*-value of less than 0.05 was considered statistically significant.

## 3. Results

A total of 57 women were initially assessed for eligibility, and seven individuals were excluded due to screening laboratory findings outside the normal range (n = 4) or a level of FSH lower than 30 IU/L (n = 3). Fifty postmenopausal women were randomly allocated to receive either the GEN product or the PLA product (Figure 1). The mean age of the participants was 55.8 ± 3.4 years (range: 48–64 years). The average FSH of the participants was 74.6 ± 20.9 IU/L (range: 42–135 IU/L). The participants’ baseline characteristics are shown in Table 1.

No participants withdrew or were withdrawn from the trial throughout the study period. Five individuals (two in the GEN group and three in the PLA group) used the product less than 50% of the expected dosage and were defined as poorly compliant. Therefore, the ITT approach included 50 participants (25 in the GEN group and 25 in the PLA group) in the analysis, while the PP approach included 45 participants (23 in the GEN group and 22 in the PLA group) in the analysis (Figure 1).

Analysis of the percentage mean changes in skin wrinkling parameters and skin color parameters found no significant difference between the two groups across all the parameters, except for skin redness, which was significantly higher in the GEN group than in the PLA group in both the ITT and PP analyses (ITT: 6.89 ± 20.94% vs. −2.97 ± 11.02%, *p* = 0.044; PP: 9.35 ± 19.74% vs. −3.90 ± 11.37%, *p* = 0.009) (Table 2).

Skin hydration was significantly increased by 12.30 ± 15.66% (95% CI, 6.16–18.44) and 7.90 ± 15.54% (95% CI, 1.80–13.99) following the application of the GEN product and the PLA product, respectively. However, no significant difference in the percentage mean change in skin hydration was found between the two groups in both ITT and PP analyses (ITT: MD = 4.40%, 95% CI, −4.47–13.28, *p* = 0.323; PP: MD = 5.84%, 95% CI, −3.81–15.48, *p* = 0.229).

For facial skin analysis, fine pores and their area were significantly decreased by −2.92 ± 7.19 pores (95% CI, −5.74–−0.10) and −0.41 ± 0.79% (95% CI, −0.72–−0.10), respectively, following the application of the GEN product, while no significant change was observed in the PLA group. There were no significant changes in other facial skin parameters following the application of either the GEN product or the PLA product. Also, no significant difference in any facial skin parameters between the two groups was observed in both ITT and PP analyses (Table 3). No significant difference between the two groups was found in the fluorescing acne lesions in terms of either size or quantity (Table 3).

Subgroup analysis of older women (age ≥ 56 years) with adequate compliance found significant differences between the two groups in the percentage mean changes of most skin wrinkling parameters, including skin roughness (−11.41 ± 13.92% vs. 0.39 ± 14.97%, *p* = 0.048), average roughness (−11.99 ± 16.36% vs. 1.68 ± 15.55%, *p* = 0.040), smooth depth (−12.14 ± 18.87% vs. 4.45 ± 18.92%, *p* = 0.035), arithmetic average roughness (−15.74 ± 16.12% vs. 4.37 ± 15.79%, *p* = 0.004), and surface (−7.13 ± 12.73% vs. 4.00 ± 9.73%, *p* = 0.021) (Figure 2).

Most skin wrinkling parameters were significantly decreased following the application of the GEN product (skin roughness: −11.41%, 95% CI, −18.70–−4.12%; maximum roughness: −12.05%, 95% CI, −20.35–−3.74%; average roughness: −11.99%, −20.56–−3.41%; smooth depth: −12.14%, 95% CI, −22.03–−2.26%; arithmetic average roughness: −15.74%, 95% CI, −24.19–−7.30%; surface: −7.13%, −13.80 to −0.46%), whereas none of such parameters were significantly changed following the application of the PLA product (Figure 3).

The GEN product received a higher participants’ satisfaction in three aspects: wrinkle reduction (ITT: 9.36 ± 0.64 vs. 8.56 ± 1.64, *p* = 0.030; PP: 9.39 ± 0.66 vs. 8.50 ± 1.74, *p* = 0.033), overall efficacy (ITT: 9.64 ± 0.57 vs. 8.88 ± 1.33, *p* = 0.013; PP: 9.61 ± 0.58 vs. 8.86 ± 1.42, *p* = 0.031), and serum formulation (ITT: 9.72 ± 0.54 vs. 9.16 ± 1.03, *p* = 0.021; PP: 9.70 ± 0.56 vs. 9.18 ± 1.10, *p* = 0.058) (Appendix A).

During the six-week study period, only one individual in the GEN group experienced mild skin rashes on both cheeks on Day 5, which resolved after oral antihistamine therapy for 2 days (Days 6–7). The GEN product was withheld for 3 days (Days 6–8) until completely resolved and continued after that until the end of the study. No skin rash occurred again after re-exposure to the GEN product.

## 4. Discussion

The present study demonstrated that the combination of genistein, vitamin E, vitamin B3, and ceramide in a topical skincare product could improve certain features of facial skin appearance in postmenopausal women. This clinical trial was executed to the gold standard, with a randomized, double-blind, placebo-controlled design in which study creams were coded and randomly assigned at the source, while the research participants and the investigators were kept blind about the coding until study completion. Facial skin outcomes (including skin wrinkles, color, hydration, pores, spots, and acne) were measured objectively and quantitatively using multiple well-established tools.

Our analysis revealed that postmenopausal women, particularly those who are older, could benefit from the GEN product in terms of skin wrinkles. It is well known that genistein and 17-β-estradiol share some similarities in their chemical structure, and genistein is characterized as a partial agonist for the beta isoform of the estrogen receptor (ERβ) [9,21,22]. In other words, as a partial agonist, genistein may behave differently on the ERβ, depending on whether the estrogen environment is low or high [23,24]. Therefore, our observation is reasonable since older postmenopausal women who live through a longer duration of estrogen deficiency after their last menstruation tend to have lower estradiol levels than those at a younger age [23,25]. In addition, our results are supported by the fact that the topical application of 4% genistein has been shown to increase skin thickness and blood vessels [26], the concentration of hyaluronic acid, glycosaminoglycan, and fibroblasts in the dermis [27], as well as the amount of type I and type III facial collagen [17], all of which are factors leading to a reduction in skin wrinkles. Moreover, apart from genistein, the GEN product contained some other active substances, including vitamin E [12,28], vitamin B3 [19,29], and ceramide [20,30], all of which have evidence indicating their anti-wrinkle properties [31].

We also observed an increase in skin redness among those who applied the GEN product when compared to the others. Since skin color changes may provide some clues to intervention responses, it is important to consider the sources of variance associated with the individuals, the environment, and the colorimetric principles. Melanin, hemoglobin, carotene, and bilirubin are the main chromophores that determine skin color [32], and their concentrations may vary depending on the individual’s phenotype, age, anatomic location, external insults of chemical irritants and ultraviolet radiation, and physiological and mental conditions [33,34,35]. In our study, the skin redness pattern in the GEN group was unlikely to be inflammatory erythema, which typically presents a far lower value of skin redness than that shown in our participants [36,37]. Rather, it is more likely to be associated with increased vasodilation and vascularization of the skin, as manifested by oxygenated blood coloration, without any signs of inflammation [33,38,39]. Our finding is consistent with the previous evidence indicating that genistein is able to improve mitochondrial membrane potential and increase nitric oxide release under physiological conditions [40]. In addition, genistein may increase dermal collagen, elastin, and the number of dermal blood vessels [41]. The literature suggests that enhanced skin redness following topical use of the GEN product appears to be an indicator of postmenopausal women having healthy facial skin. However, an increase in skin redness following the GEN product application is also worth noting since postmenopausal women with some underlying conditions, such as rosacea, may experience difficulties applying the GEN product. For example, it is quite common that hot flashes that come with menopause may exacerbate rosacea signs and symptoms, causing rosacea to flare up [42]. Therefore, postmenopausal women with such underlying conditions may become worried if their facial skin becomes redder after using the GEN product. It is worth noting that skin hydration was improved, while fine pores and their area were reduced, following topical application of either the GEN product or the PLA product. It is plausible to assume that these effects would have been attributable to the cream base found in both products, which contained skin moisturizing substances (e.g., fats and humectants) and excipients (e.g., emulsifiers, chelating reagents, and preservatives) [43]. Notwithstanding that the analysis found no statistically significant difference between the two groups, the incorporation of genistein, vitamin E, vitamin B3, and ceramide into the cream base tended to produce more effects on skin hydration than the cream base alone. When the facial skin around the pores is properly hydrated, the stratum corneum inside and surrounding the pores swells, and the size of the pores and their surface area are reduced accordingly [44]. Since skin elasticity is partly influenced by skin hydration levels at viable epidermis depths and skin roughness at the stratum corneum, it is conceivable that the GEN product may lead to enhanced skin elasticity as well [45]. However, the present study had no parameters representative of skin elasticity, so the benefit of the GEN product on this feature may require further investigation.

The present study demonstrated that the GEN product, when topically applied for 6 weeks, was considered safe. Only one participant experienced mild skin rashes on both cheeks, possibly owing to vigorous scrubbing and over-exfoliating the skin. This adverse event was unlikely to be associated with the GEN product because it did not occur again after a rechallenge with the GEN product.

The results of the present study should be interpreted in light of the study’s context and limitations. First, the intervention was evaluated over a six-week duration in postmenopausal women. Therefore, the findings do not provide information about applications that would last for a longer period of time. Some benefits of the GEN product might have been seen if the study had been undertaken for a longer duration [17,26,27,46]. Second, subgroup analysis was conducted based on the age of the research participants instead of the number of years after they ceased to menstruate. Although the latter variable appears to be more logical in relation to the estradiol levels, we were unable to determine it due to a lack of data indicating the date/year of the last menstrual period. Such data should be collected in future studies examining the effect of an intervention aimed at addressing conditions related to estrogen deficiencies. Third, we did not examine the benefits of the GEN product on facial skin at the molecular level. Further research into the advantages of the GEN product using cutting-edge molecular testing and/or some more recent technologies, such as artificial intelligence-based methods for assessing facial skin lesions and conditions, may be of interest [47,48].

## 5. Conclusions

The GEN product has advantages for postmenopausal women’s facial skin, especially for those who are older. It can moisturize facial skin while also reducing wrinkles and enhancing redness. The GEN product is safe and achieves high levels of participant satisfaction with regard to product usage. Further research is required to close the gap identified in this study, and it ought to be conducted over a longer duration with more research participants. It may be beneficial to investigate further how different genistein-based formulations might affect the product’s efficacy when combined with other skin-lightening and/or antioxidant ingredients.

## Figures and Tables

**Figure 1 jcm-12-01326-f001:**
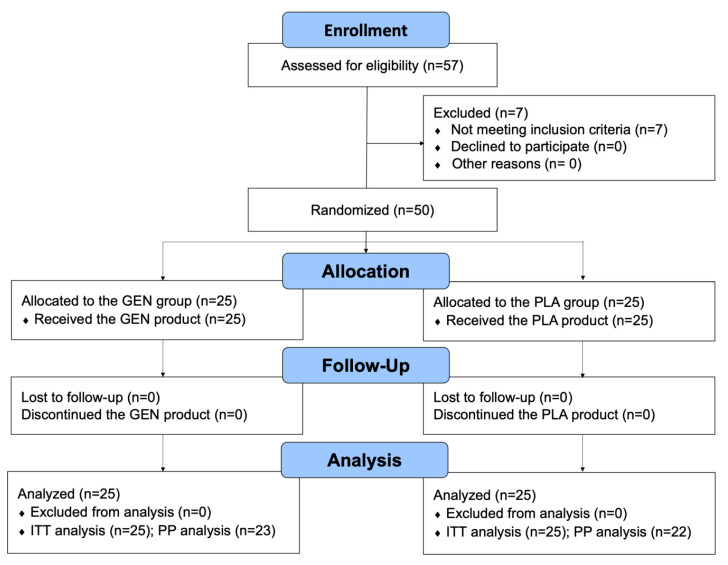
Consort flow diagram.

**Figure 2 jcm-12-01326-f002:**
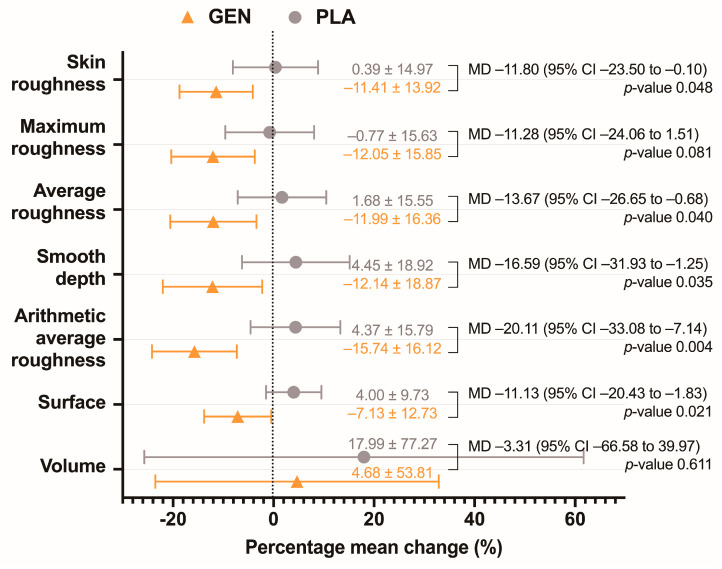
Percentage mean changes of skin wrinkling parameters in older women (age ≥ 56 years). An orange triangle represents the GEN group, while a grey circle represents the PLA group. Abbreviation: MD, mean difference; CI, confidence interval.

**Figure 3 jcm-12-01326-f003:**
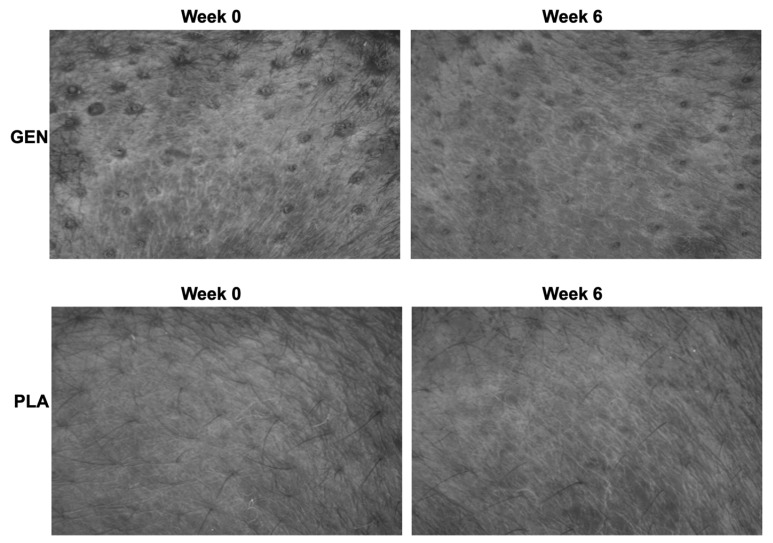
Replica image of the topography of the skin surface at baseline and week 6. Abbreviation: GEN, the Genistein Nutraceutical product; PLA, placebo.

**Table 1 jcm-12-01326-t001:** The participants’ baseline characteristics.

Parameters	GEN Group (n = 25)	PLA Group (n = 25)
Demographics		
Age (years)	55.8 ± 3.8	55.8 ± 3.0
BMI (kg/m^2^)	24.0 ± 2.2	23.7 ± 3.1
FSH (IU/L)	75.2 ± 23.4	74.0 ± 18.6
Skin wrinkle parameters		
Skin roughness (arb. unit)	59.01 ± 9.65	54.55 ± 8.46
Maximum roughness (arb. unit)	49.76 ± 9.12	45.08 ± 8.44
Average roughness (arb. unit)	38.56 ± 6.98	34.85 ± 6.57
Smooth depth (arb. unit)	30.12 ± 5.96	27.39 ± 4.98
Arithmetic average roughness (arb. unit)	8.76 ± 1.69	7.96 ± 1.56
Surface (%)	402.15 ± 52.07	374.36 ± 45.04
Volume (mm^2^)	55.08 ± 20.00	53.91 ± 17.90
Skin color parameters		
Brightness (arb. unit)	60.33 ± 3.97	59.56 ± 2.17
Redness (arb. unit)	13.33 ± 1.67	13.44 ± 1.76
Pigmentation (arb. unit)	16.41 ± 1.92	17.03 ± 1.65
Individual typology angle (arb. unit)	31.48 ± 10.91	29.29 ± 6.47
Skin hydration parameters		
Hydration value (arb. unit)	78.10 ± 10.21	78.87 ± 7.43
Facial skin parameters		
Fine pore (arb. unit)	22.48 ± 15.59	20.96 ± 11.85
Fine pore area (%)	2.32 ± 1.69	2.10 ± 1.21
Large pore (arb. unit)	6.88 ± 9.55	7.76 ± 7.50
Pore area (%)	1.62 ± 2.03	1.87 ± 1.75
Spot (arb. unit)	0.80 ± 1.04	1.08 ± 0.91
Spot area (%)	0.79 ± 1.17	1.30 ± 1.39
Acne parameters		
Size (%)	0.55 ± 1.69	0.42 ± 0.75
Quantity (arb. unit)	5.87 ± 10.59	6.93 ± 9.31

**Table 2 jcm-12-01326-t002:** Percentage mean changes in skin wrinkling, skin color, and hydration parameters.

Parameters	Intention-to-Treat Analysis			Per-Protocol Analysis		
	GEN Group (n = 25)	PLA Group (n = 25)	*p*-Value	GEN Group (n = 23)	PLA Group (n = 22)	*p*-Value
Skin wrinkle parameters						
Skin roughness	−4.80 ± 16.82	−3.13 ± 14.14	0.704	−5.46 ± 17.03	−2.99 ± 15.08	0.610
Maximum roughness	−4.78 ± 17.48	−2.86 ± 14.34	0.673	−5.69 ± 17.51	−3.25 ± 15.22	0.621
Average roughness	−4.00 ± 18.85	−1.06 ± 13.67	0.531	−4.97 ± 19.15	−1.17 ± 14.51	0.459
Smooth depth	−2.29 ± 22.82	−1.25 ± 18.44	0.549	−4.20 ± 20.79	1.75 ± 19.61	0.329
Arithmetic average roughness	−3.17 ± 22.18	−0.12 ± 16.46	0.584	−4.44 ± 21.35	0.51 ± 17.31	0.399
Surface	−1.74 ± 16.03	2.16 ± 8.55	0.291	−2.66 ± 16.40	2.30 ± 9.05	0.215
Volume	17.41 ± 59.50	15.59 ± 70.35	0.922	17.23 ± 61.74	20.44 ± 73.64	0.875
Skin color parameters						
Brightness	−0.14 ± 10.12	2.31 ± 4.62	0.280	−1.35 ± 9.59	2.52 ± 4.74	0.094
Redness	6.89 ± 20.94	−2.97 ± 11.02	0.044	9.35 ± 19.74	−3.90 ± 11.37	0.009
Pigmentation	−3.78 ± 19.34	−5.40 ± 11.13	0.718	−4.56 ± 17.88	−5.34 ± 11.62	0.863
Individual typology angle	35.68 ± 141.42	19.81 ± 28.92	0.587	27.60 ± 142.14	21.46 ± 29.81	0.844
Skin hydration parameters						
Hydration value	12.30 ± 15.66	7.90 ± 15.54	0.323	13.23 ± 16.01	7.39 ± 16.06	0.229

**Table 3 jcm-12-01326-t003:** Absolute mean changes in facial skin and acne parameters.

Parameters	Intention-to-Treat Analysis			Per-Protocol Analysis		
	GEN Group (n = 25)	PLA Group (n = 25)	*p*-Value	GEN Group (n = 23)	PLA Group (n = 22)	*p*-Value
Facial skin parameters						
Fine pore	−2.92 ± 7.19	−0.48 ± 7.57	0.248	−3.39 ± 7.18	−0.82 ± 7.10	0.234
Fine pore area	−0.41 ± 0.79	−0.03 ± 0.78	0.093	−0.47 ± 0.79	−0.06 ± 0.69	0.075
Large pore	−0.84 ± 6.42	−0.04 ± 3.62	0.590	−1.00 ± 6.67	0.00 ± 3.82	0.543
Large pore area	−0.18 ± 1.42	0.13 ± 0.98	0.368	−0.21 ± 1.48	0.14 ± 1.02	0.356
Spot	−0.20 ± 0.82	−0.04 ± 0.68	0.454	−0.26 ± 0.69	−0.14 ± 0.64	0.534
Spot area	−0.28 ± 1.12	−0.11 ± 0.71	0.534	−0.38 ± 1.01	−0.16 ± 0.72	0.395
Acne parameters						
Size	−0.13 ± 0.46	−0.17 ± 0.52	0.786	−0.14 ± 0.48	−0.16 ± 0.55	0.927
Quantity	−0.97 ± 5.19	−1.93 ± 5.10	0.512	−1.51 ± 4.75	−1.91 ± 5.43	0.792

## Data Availability

All data used to support the findings of this study are available from the corresponding author upon reasonable request.

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
