# Peer review of "Efficacy and Safety of the Genistein Nutraceutical Product Containing Vitamin E, Vitamin B3, and Ceramide on Skin Health in Postmenopausal Women: A Randomized, Double-Blind, Placebo-Controlled Clinical Trial"

_jcm, 2023, doi:10.3390/jcm12041326_

Round 1
Reviewer 1 Report
I suggest change the title to: Efficacy and safety of a product containing genistein nutraceutical, ceramide and vitamins E and B3 on skin health in postmenopausal woman: a randomized, double-bind, placebo-controled clinical trial.
Please add the amount of vitamins E and B3 and of ceramide in the product utilized in the research
Author Response
We thank the editors and reviewers for the careful and insightful review of our manuscript. We sincerely appreciate all the valuable comments and suggestions, which help us improve the quality of our manuscript. Our responses to the comments are described below in a point-by-point manner. Appropriate changes, suggested by the reviewers, have been added to the manuscript. In any case, we are open to consideration of any further comments. Thank you for your consideration.
Reviewer 1’s comment
I suggest change the title to: Efficacy and safety of a product containing genistein nutraceutical, ceramide and vitamins E and B3 on skin health in postmenopausal woman: a randomized, double-bind, placebo-controled clinical trial.
Author’s response:
Thank you for your suggestion. According to the reviewer’s suggestion, we have changed the title to “Efficacy and Safety of the Genistein Nutraceutical Product Containing Vitamin E, Vitamin B3, and Ceramide on Skin Health in Postmenopausal Women: A Randomized, Double-Blind, Placebo-Controlled Clinical Trial.”
Reviewer 1’s comment
Please add the amount of vitamins E and B3 and of ceramide in the product utilized in the research.
Author’s response:
We have added the amount of vitamin E and B3 and of ceramide in the product utilized in the research. It now reads “The Genistein Nutraceutical (GEN) product consisted of 4% genistein, 1% vitamin E, 1% vitamin B3, and 0.2% ceramide.” (Page 3, Line 119)
Reviewer 2 Report
The authors describe the application study of the genistein-based formulation in an interesting and expert manner. I believe that the study group could have been larger. The only question that comes to my mind is about the continuation of the study. In the conclusion it is written that the study should be expanded and continued, I would ask you to specify in what direction this should go?
Author Response
We thank the editors and reviewers for the careful and insightful review of our manuscript. We sincerely appreciate all the valuable comments and suggestions, which help us improve the quality of our manuscript. Our responses to the comments are described below in a point-by-point manner. Appropriate changes, suggested by the reviewers, have been added to the manuscript. In any case, we are open to consideration of any further comments. Thank you for your consideration.
Reviewer 2’s comment
The authors describe the application study of the genistein-based formulation in an interesting and expert manner. I believe that the study group could have been larger. The only question that comes to my mind is about the continuation of the study. In the conclusion it is written that the study should be expanded and continued, I would ask you to specify in what direction this should go?
Author’s response:
Thank you for your comments. According to the reviewer’s comment, we have added more descriptions in the conclusions section. It now reads “Further research is required to close the gap identified in this study, and it ought to be conducted over a longer duration with more research participants. It may be beneficial to investigate further how different genistein-based formulations might affect the product’s efficacy when combined with other skin-lightening and/or anti-oxidant ingredients.” (Page 11, Lines 348-350)
Reviewer 3 Report
In the manuscript entitled “Efficacy and Safety of the Genistein Nutraceutical Product on Skin Health in Postmenopausal Women: A Randomized, Double-Blind, Placebo-Controlled Clinical Trial” from Takuathung et al, the authors explored the efficacy and safety of a genistein-based product. The paper fits the Journal scope, the work is interesting and is well written. I have just some comments:
1. Which alterations on the formulation can be proposed to improve the studied parameters?
2. The authors refer that evaluated the participant satisfaction of this product. Did they collect information of their opinion regarding the smell, color and texture of the product?
3. Figure 3 is not correct. Please, replace it.
4. Please, remove the last two sentences of the section “Results”
5. “It is well known that genistein has a chemical structure similar to 17-β-estradiol”. I don’t agree with this sentence… 17-β-estradiol has a steroidal structure, different of genistein structure. Maybe the functional groups OH are “strategically” (in space) positioned to interact with the estrogen receptor…
6. “genistein can act as either an ERβ agonist or an ERβ antagonist in the absence or presence 272 of ER’s endogenous hormone, 17-β-estradiol”. Please, confirm this affirmation. An agonist induces alteration of the conformation of the receptor, opposed to an antagonist.
Author Response
We thank the editors and reviewers for the careful and insightful review of our manuscript. We sincerely appreciate all the valuable comments and suggestions, which help us improve the quality of our manuscript. Our responses to the comments are described below in a point-by-point manner. Appropriate changes, suggested by the reviewers, have been added to the manuscript. In any case, we are open to consideration of any further comments. Thank you for your consideration.
Reviewer 3’s comment
In the manuscript entitled “Efficacy and Safety of the Genistein Nutraceutical Product on Skin Health in Postmenopausal Women: A Randomized, Double-Blind, Placebo-Controlled Clinical Trial” from Takuathung et al, the authors explored the efficacy and safety of a genistein-based product. The paper fits the Journal scope, the work is interesting and is well written. I have just some comments:
1. Which alterations on the formulation can be proposed to improve the studied parameters?
Author’s response:
Thank you for your comments and suggestions. We are considering looking into various combinations with additional skin-lightening and/or antioxidant substances. According to the reviewer’s comment, we have added more descriptions in the conclusions section. It now reads “Further research is required to close the gap identified in this study, and it ought to be conducted over a longer duration with more research participants. It may be beneficial to investigate further how different genistein-based formulations might affect the product’s efficacy when combined with other skin lightening and/or anti-oxidant ingredients.” (Page 11, Lines 348-350)
Reviewer 3’s comment
2. The authors refer that evaluated the participant satisfaction of this product. Did they collect information of their opinion regarding the smell, color and texture of the product?
Author’s response:
The participant’s satisfaction with the product was assessed using the questionnaire which included six domains: serum formulation, wrinkle reduction, hydration, overall efficacy, safety profile, and whether to recommend the product to friends and family (Table S1). Although there were no specific questions regarding the smell, color, or texture of the product, the questionnaire included a free space for participants to express their opinion. General comments from the participants included the gentle scent and delicate touch of the product. None of the comments mentioned the product’s color.
Reviewer 3’s comment
3. Figure 3 is not correct. Please, replace it.
Author’s response:
Thank you very much for your notification. We have replaced it with the correct figure. (Page 9)
Reviewer 3’s comment
4. Please, remove the last two sentences of the section “Results”
Author’s response:
We do apologize for this mistake, forgetting to remove the default text in the Journal’s template. In the revised manuscript, we have corrected it.
Reviewer 3’s comment
5. “It is well known that genistein has a chemical structure similar to 17-β-estradiol”. I don’t agree with this sentence… 17-β-estradiol has a steroidal structure, different of genistein structure. Maybe the functional groups OH are “strategically” (in space) positioned to interact with the estrogen receptor…
Author’s response:
Thank you very much for your careful consideration. According to the reviewer’s comment, we have revised the statement to make the message clearer and more accurate. In the revised manuscript, it now reads “genistein and 17-β-estradiol share some similarities in their chemical structure, and genistein is characterized as a partial agonist for the beta isoform of the estrogen receptor (ERβ).” (Page 9, Lines 273-275)
Reviewer 3’s comment
6. “genistein can act as either an ERβ agonist or an ERβ antagonist in the absence or presence 272 of ER’s endogenous hormone, 17-β-estradiol”. Please, confirm this affirmation. An agonist induces alteration of the conformation of the receptor, opposed to an antagonist.
Author’s response:
Thank you for your careful consideration. According to the reviewer’s comment, we have reconsidered that statement and revised it accordingly. In the revised manuscript, it now reads “In other words, as a partial agonist, genistein may behave differently on the ERβ, depending on whether the estrogen environment is low or high.” (Page 9, Lines 275-277)